# The Alternating Access Mechanism in Mammalian Multidrug Resistance Transporters and Their Bacterial Homologs

**DOI:** 10.3390/membranes13060568

**Published:** 2023-05-30

**Authors:** Shadi A Badiee, Ugochi H. Isu, Ehsaneh Khodadadi, Mahmoud Moradi

**Affiliations:** Department of Chemistry and Biochemistry, University of Arkansas, Fayetteville, AR 72701, USA; sasadian@uark.edu (S.A.B.); uhisu@uark.edu (U.H.I.); ekhodada@uark.edu (E.K.)

**Keywords:** multidrug resistance, MDR, ATP-binding cassette transporter, ABC transporters, lipid flippase, MsbA, Sav1866, multidrug resistance protein, MRP1, Pgp, P-glycoprotein, nucleotide-binding domains, NBDs, transmembrane domains, TMDs, alternating access mechanism, alternating access model

## Abstract

Multidrug resistance (MDR) proteins belonging to the ATP-Binding Cassette (ABC) transporter group play a crucial role in the export of cytotoxic drugs across cell membranes. These proteins are particularly fascinating due to their ability to confer drug resistance, which subsequently leads to the failure of therapeutic interventions and hinders successful treatments. One key mechanism by which multidrug resistance (MDR) proteins carry out their transport function is through alternating access. This mechanism involves intricate conformational changes that enable the binding and transport of substrates across cellular membranes. In this extensive review, we provide an overview of ABC transporters, including their classifications and structural similarities. We focus specifically on well-known mammalian multidrug resistance proteins such as MRP1 and Pgp (MDR1), as well as bacterial counterparts such as Sav1866 and lipid flippase MsbA. By exploring the structural and functional features of these MDR proteins, we shed light on the roles of their nucleotide-binding domains (NBDs) and transmembrane domains (TMDs) in the transport process. Notably, while the structures of NBDs in prokaryotic ABC proteins, such as Sav1866, MsbA, and mammalian Pgp, are identical, MRP1 exhibits distinct characteristics in its NBDs. Our review also emphasizes the importance of two ATP molecules for the formation of an interface between the two binding sites of NBD domains across all these transporters. ATP hydrolysis occurs following substrate transport and is vital for recycling the transporters in subsequent cycles of substrate transportation. Specifically, among the studied transporters, only NBD2 in MRP1 possesses the ability to hydrolyze ATP, while both NBDs of Pgp, Sav1866, and MsbA are capable of carrying out this reaction. Furthermore, we highlight recent advancements in the study of MDR proteins and the alternating access mechanism. We discuss the experimental and computational approaches utilized to investigate the structure and dynamics of MDR proteins, providing valuable insights into their conformational changes and substrate transport. This review not only contributes to an enhanced understanding of multidrug resistance proteins but also holds immense potential for guiding future research and facilitating the development of effective strategies to overcome multidrug resistance, thus improving therapeutic interventions.

## 1. Introduction

Multidrug resistance (MDR) is one of the obstacles in modern medicine since it reduces the efficacy of several drugs and makes treatment less successful. Resistance to antibiotics, anticancer, and antiviral treatments are mediated by these proteins, which are classified as one of the subfamilies in ATP-binding cassette (ABC) transporters. ABC transporters are composed of two distinct parts called the nucleotide-binding domain (NBD) and the transmembrane domain (TMD). The NBD provides energy for the movement of substrates in or out of the cell through having a binding site for ATP, whereas TMD supplies the substrate translocation route. The selective efflux of substrates is made possible by the alternating access mechanism, in which the transporter switches between inward- and outward-facing conformations.

P-glycoprotein (Pgp, also referred to as MDR1 and ABCB1) and Multidrug resistance protein 1 (MRP1) are critical multidrug resistance in mammalian cancer cells, and these transporters can efflux several different types of medicines. MsbA and Sav1866 are bacterial homologs of mammalian Pgp. Based on their TMD fold, these four transporters are all classified as group IV. Although the TMDs are arranged similarly, their features, such as NBDs, ATP hydrolysis, NBD-TMD interface, transporter mechanism details, substrate specificity, substrate binding pocket, etc., may differ, case by case. In this review paper, we attempt to describe the majority of these features using experimental and computational papers. The transporters MRP1, Pgp, Sav1866, and MsbA were chosen for this review paper to include both bacterial (Sav1866, MsbA) and mammalian (Pgp, MRP1) transporters.

Several recent review papers have thoroughly covered various aspects of the ABC superfamily, such as the role of ABC transporters in specific cancer progression and metastasis, the development of novel ABC transporter modulators, and the collection of data on natural and synthetic compounds targeting cancer-related ABC transporters, among others [1,2,3,4,5,6,7,8,9]. Our review paper, however, specifically focuses on mammalian proteins and their bacterial homologs, aiming to comprehensively gather data on all aspects of these proteins, particularly their alternating access mechanisms. In this review paper, we briefly explain the variety of efflux pumps, highlighting their importance in multidrug resistance (Section 1.1). This is followed by a broad overview of the ATP-binding cassette (ABC) superfamily (Section 1.2), specifically focusing on the multidrug resistance (MDR) subfamily and their general structure and function (Section 1.3). We also discuss their alternating access mechanisms (Section 1.4) and highlight two vital MDR proteins (MRP1 and Pgp) and the homologs of Pgp (Sav1866, MsbA), providing a detailed analysis of their structures and mechanisms (Section 2).

### 1.1. Efflux Pumps

Efflux pumps are membrane proteins that play a crucial role in removing various compounds from cells [10,11,12]. Efflux pumps are found in diverse organisms, encompassing both prokaryotes and eukaryotes. However, there are variations in the types and characteristics of efflux pumps between these two groups. Although certain classes of efflux pumps can confer resistance in both prokaryotic and eukaryotic organisms, there are notable differences in their specific functions and mechanisms [5,13]. Prokaryotic efflux pumps are classified into six distinct groups, whereas eukaryotic efflux pumps are classified into five groups. Efflux pumps play crucial roles in a multitude of cellular processes. These include maintaining the proper potential and pH gradient across the cell membrane, participating in intercellular signaling, contributing to microbial virulence, and facilitating the elimination of unwanted metabolites and toxic substances from the cell. Through these functions, efflux pumps actively contribute to the maintenance of cellular homeostasis [10]. Efflux pumps can be classified into five primary families, namely small multidrug resistance (SMR), multi-antimicrobial extrusion (MATE), major facilitator superfamily (MFS), resistance nodulation and cell division (RND), and ATP-binding cassette (ABC). The ABC family transporters harness energy obtained from ATP hydrolysis. In contrast, pumps belonging to the other four families operate as second-order transporters, utilizing a proton gradient for their transport function [10]. Although efflux pumps are commonly linked to multidrug resistance, the ABC efflux pumps stand out as the primary players involved in multidrug resistance [5], with other efflux pumps possessing unique roles and functions extending beyond MDR [14,15]. The structures and roles of the other types of efflux pumps are briefly described below.

#### 1.1.1. Major Facilitator Superfamily

MFS efflux pumps are membrane proteins that actively transport a wide range of substrates across cellular membranes [16]. Examples include Escherichia coli MdfA and TetA [17,18]. As part of the largest superfamily of secondary active transporters, MFS efflux pumps utilize the energy from electrochemical gradients to transport substrates bidirectionally [19]. They play a role in maintaining cellular homeostasis and regulating molecule levels within the cell. These pumps are diverse and can transport organic compounds, ions, and drugs out of the cell, aiding in cellular detoxification [19]. In some bacteria, MFS efflux pumps are associated with multidrug resistance by expelling antibiotics and contributing to drug resistance [19]. The substrate-binding site in MFS transporters is typically located in a central cavity, involving residues from both the N- and C-terminal domains. Multidrug recognition is achieved through a large substrate-binding pocket that can accommodate compounds of different sizes and shapes [17,19].

#### 1.1.2. Multi-Antimicrobial Extrusion

MATE is a diverse family of secondary active transporters [20] composed of 12 transmembrane helices [21]. These transporters are found in all three domains of life [20] and play a crucial role in facilitating the transport of a wide range of compounds across cellular and organellar membranes [22]. Bacterial MATE transporters have a specific affinity for cationic compounds, including important antibiotics. Their presence contributes significantly to the multi-drug resistance observed in pathogenic bacteria, as these transporters actively extrude antibiotics and confer resistance to these drugs [20]. In plants, MATE transporters are involved in the accumulation of various metabolites within organelles, particularly vacuoles. They play a vital role in maintaining cellular homeostasis by actively transporting specific compounds to their designated locations [23]. In humans, MATE transporters are expressed in the brush-border membrane of the kidney. They are responsible for the clearance of cationic drugs from the body, ensuring their elimination through urine [20]. MATE transporters can be classified into different subfamilies based on their amino acid sequence similarity. These subfamilies include NorM, DinF (DNA-damage inducible protein F), and eukaryotic MATE transporters [24,25].

#### 1.1.3. Small Multidrug Resistance

SMR efflux pumps, which are part of the small multidrug resistance (SMR) family, consist of four transmembrane helices and typically form homodimers or sometimes heterodimers [12,25]. These pumps play a crucial role in exporting a diverse range of toxic compounds from bacterial cells. Their compact size, with a short polypeptide length of 100–150 amino acids [26], and the presence of short hydrophilic loops enable them to solubilize and transport various drugs [12]. Examples of SMR efflux pumps include EmrE, QacG, and QacH, found in different bacterial species [27]. These pumps actively export toxic compounds such as quaternary ammonium compounds and dyes, protecting the cells from their harmful effects [25]. By facilitating the removal of these substances, SMR efflux pumps contribute to bacterial resistance against a wide range of compounds [12,25,26].

#### 1.1.4. Resistance-Nodulation-Division

RND efflux pumps form tripartite complexes composed of three components: an inner membrane protein, a periplasmic membrane fusion protein, and an outer membrane protein [28,29]. Together, these complexes facilitate the active efflux of various substrates from bacterial cells. Examples of RND efflux pumps include AcrAB-TolC and MexAB-OprM, found in Gram-negative bacteria such as *E. coli* and *Pseudomonas aeruginosa*, respectively [28,30]. These pumps are known for their ability to actively pump out a diverse range of substrates, including antibiotics, detergents, and heavy metals. By actively removing these substances from the cell, RND efflux pumps play a crucial role in bacterial resistance, allowing the cells to survive and adapt in different environments [25,28]. They serve as a defense mechanism by protecting the cell from harmful compounds and are also involved in the development of multidrug resistance. The structure of RND transporters is distinct from other secondary transporters, characterized by a conserved transmembrane region and a periplasmic and luminal domain that contribute significantly to the protein’s mass [31]. The transmembrane region contains an internal structural repeat resulting from gene duplication, while the periplasmic luminal domain can vary among different RND homologs [25].

### 1.2. ABC Transporter

ATP-binding cassette (ABC) transporters are a vast class of membrane proteins that utilize the energy from ATP hydrolysis (bond breakage between γ and β phosphate in ATP) to carry a number of different substrates such as peptides [32], ions, lipids, vitamins, amino acids, drugs, etc., across the membrane [33,34]. They can be homo or heterodimers comprising two conserved cytoplasmic NBDs and two distinct TMDs [35,36,37,38]. NBDs act as gatekeepers and are necessary to pass molecules through the membrane bilayer. In addition to binding and hydrolyzing ATP, they can induce conformational changes in the related TMDs. This alteration allows substrates to traverse the membrane bilayer and pass into or out of the cytoplasm [39,40]. It is known that both prokaryotic and eukaryotic/mammalian organisms are members of the ABC transporter superfamily [41].

ABC transporters in humans are classified into five subfamilies based on their sequence: ABCA, ABCB, ABCC, ABCD, and ABCG [42,43,44,45]. There are two types of ABC transporters based on the direction of substrate movement: importers and exporters. Eukaryotes, unlike bacteria that can yield both ABC importers and exporters, have mainly exporters with some exceptions [46]. ABC transporters were also historically divided into three groups: exporters, importers, and non-transporters [47,48]. However, a recent paper by Thomas et al. (2020) [42] proposed a new classification based on the structural homology of TMDs. This new classification of ABC transporters is implemented in response to the wealth of high-resolution structural information available from X-ray crystallography and cryo-electron microscopy. These methods uncovered diverse TMD folds and evolutionary relationships between bacterial and mammalian transporters [49,50,51,52,53]. By integrating this new structural knowledge with previous phylogenetic analyses, a new classification has been introduced. In accordance with this novel classification, ABC transporters can be divided into seven categories (I–VII) based solely on their TMD fold without considering any additional membrane-integrated domains. TMD fold is determined by utilizing TM-scores obtained from comparing the structural alignment of the TMDs. For instance, Class IV (Figure 1) represents a TMD fold with a 6 + 6 transmembrane helix organization. It can exist as a homo- or heterodimer within a single chain. Notably, transporters such as Pgp, MsbA, Sav1866, and MRP1 fall into this class [42].

Table 1 shows a list of available PDB files for Pgp, MRP1, Sav1866, and MsbA transporter proteins. These files were created using advanced techniques such as X-ray crystallography and cryo-electron microscopy. We sorted the accessible PDB files for each protein by resolution, starting with higher-resolution structures and progressing to lower-resolution structures.

### 1.3. MDRs

The ability of cells or organisms to resist the effects of numerous medications or chemotherapeutic treatments is referred to as multidrug resistance (MDR). MDR can be caused by a variety of mechanisms, including the overexpression of efflux pumps, which actively remove drugs from cells, changes in target molecules that reduce drug binding affinity, drug modification or inactivation, decreased drug uptake due to membrane transporter mutations, increased DNA repair mechanisms, and changes in signaling pathways. These mechanisms all contribute to the development of MDR and cause obstacles in disease treatment [73]. Resistance against anticancer medications is one of the significant difficulties related to developing anticancer chemotherapies [74,75]. In addition, resistance to multiple antibiotics is one of the other significant challenges regarding bacterial infections [76,77]. In cancer cells, multidrug resistance (MDR) as a phenomenon is caused by the over-expression of ABC transporters such as P-glycoprotein (Pgp, ABCB1 or MDR1), multidrug resistance-associated proteins MRPs (ABCC1), and breast cancer resistance protein BCRP (ABCG2) leading to the removal of drugs from the cell, thereby reducing their concentration below the effective level. They are expressed in a wide variety of organs, such as the liver, colon, kidney, brain, etc. [78]. MDR transporters perform critical roles, including the distribution, metabolism, and elimination of medicines, along with enabling cells to discharge anticancer or antibiotic drugs from the cell [66,79]. They have also been described as “hydrophobic vacuum cleaners” because of their capacity to remove drugs and lipids from the inner membrane leaflet [80]. Researchers have concentrated on finding inhibitors for these drug efflux pumps since they can increase the efficacy of anticancer and antibacterial medications [81].

### 1.4. NBD and TMD

Two solvent-exposed nucleotide-binding domains (NBD1 and NBD2) and two transmembrane domains (TMD1 and TMD2) make up the four functional domains of the typical ABC transporter (Figure 2A). The transmembrane domains are also known as MSD or membrane-spanning domains. Most of the eukaryotic ABC transporters are single polypeptides that contain all four domains, unlike bacterial ABC transporters, which can be composed of individual subunits or joined NBDs and TMDs. Eukaryotic ABC transporters can occasionally be built from similar or dissimilar half transporters [82]. ABC transporter structure can be divided into four groups named: full structure (NBD-TMD-NBD-TMD or TMD-NBD-TMD-NBD) [34], half structure (NBD-TMD or TMD-NBD), single structure (TMD or NBD )and ABC2 structure. ABC2 is the structure of non-transporter ABC proteins lacking TMDs. According to this system, ABC proteins in prokaryotes are either single or half structures, while those in eukaryotes can be either half or full [83].

NBD, also known as ATPase or ATP-binding sites, is a hydrophilic domain located in the cytoplasmic side of the ABC transporter and is composed of two subdomains referred to as RecA-type ATP-binding core and ABC-specific. RecA-type ATP-binding core is a central β-sheet flanked by four α-helices that participates in ATP binding and hydrolysis [42,84,85,86,87]. RecA-type ATP-binding core includes motifs or subdomains named Walker A and Walker B motifs, the Q-loop, the D-loop, and the H-switch [88,89] (Figure 2B). Walker A motif or phosphate-binding loop (P-loop) is a highly conserved sequence of amino acids, including glycine (G) and lysine (K)residues (GXXGXGKS/T), forming hydrogen bonds with the phosphate group of ATP. The threonine (T)/serine (S) residue is responsible for the interaction of nucleotide bases. Walker B motif (ϕϕϕϕDE), where ϕ shows hydrophobic residues, contains aspartic acid (D) in order to coordinate magnesium (Mg^2+^) required for ATP hydrolysis. This motif also has glutamate (E) that helps facilitate the ATP hydrolysis reaction as a general base by making water molecules more reactive. Walker A (P loop) and Walker B motifs are connected via a short loop named Q-loop which is crucial for coordinating nucleotide-binding sites (NBDs). This loop (ϕ(ϕ/Q)Q) includes glutamine residues that interact with γ-phosphate of the nucleotide [90]. Moreover, Q-loop plays a significant role in communicating with the TMDs. D-loop (SALD) connects two strands of the anti-parallel β-sheet within the RecA-like core and is involved in nucleotide binding and hydrolysis. As ATP hydrolysis happens, a conformational change occurs in a loop of amino acids known as the H-switch (switch region including conserved histidine residue), which is thought to be critical for connecting ATP hydrolysis to substrate transport. The ABC-specific subdomain is composed of three anti-parallel β-sheets known as ABC-β and an α-helical subdomain known as ABC-α. The signature motif (C-loop) is a short amino acid sequence with LSGGQ sequence [91] that is unique to ABC transporters and is situated between the Walker A and Walker B motifs, found at the end of ABC-α. The C-loop improves ATP binding stability and hydrolysis by interacting with ATP’s γ-phosphate [42]. Walker-A motif (P loop), Walker-B motif, and C-loop (signature motif) are conserved sequence motifs within the NBDs that are essential for the ATP-binding and hydrolysis capability, along with interacting between NBDs and TMDs [88,92]. In the presence of two ATP molecules, NBDs can undergo the dimerization (sandwich dimer) required to generate energy for the process. An ABC transporter can hold two ATP molecules simultaneously between Walker A and Walker B motifs in one NBD and signature motif on the other NBD subunit [86,93]. Hydrolysis of ATP causes dissociation of the nucleotide-binding sites (NBS), releasing Pi and ADP.

TMD is the hydrophobic part of the ABC transporters, which spans the membrane leaflet, and substrates have their binding sites in TMD [94]. TMDs typically consist of six transmembrane α-helices, and each functional dimer TMD has around 12 of these transmembrane segments [95], except for some MRPs (Multidrug Resistance-Associated Proteins) with 17 α-helices including 3 TMDs [96]. The TMD dimers of ABC transporters serve as translocation pathways for substrates, facilitated by specialized transmembrane helices (TMHs) with specific substructures. When substrates bind to TMDs, it triggers a series of conformational changes that enable them to traverse the membrane. The hydrolysis of ATP by NBDs provides the essential energy for the process and drives these conformational modifications. The interface between TMDs and NBDs is established by small conserved helices known as coupling helices (CHs) that interact with the catalytic domain of NBDs [66] (Figure 1).

### 1.5. Alternating Access Mechanism (AAM)

The alternating access mechanism is used by all active membrane transporters, although they might vary in structure and mechanism [97,98,99]. In this mechanism (Figure 3), the transporter proteins undergo conformational changes to cross the substrate against its concentration gradient from one side of the membrane to another. During this process, the substrate binding site of the protein is exposed inside (inward-facing, IF) or outside (outward-facing, OF) the membrane alternately (IF ↔ OF), passing through a number of potential intermediate phases [100,101,102,103], in such a way that ATP binding induces NBD dimerization and the development of the OF state. In contrast, NBD dissociation and the restoration of the IF state happens through ATP hydrolysis, resulting in the release of ADP and Pi [104,105]. Although the general structural state of the protein can remain IF or OF, various local conformational changes, such as mobility in amino acid side chains, loops, and helices, can block the substrate-binding site of these transporters [106].

## 2. Alternating Access Mechanism in Mammalian MDRs and Bacterial Homologs

This section will discuss four transporters, P-glycoprotein, Sav1866, MsbA, and MRP1, through reviewing computational and experimental papers. Pgp and MRP1 both function as multidrug resistance (MDR) proteins. Bacterial Pgp transporter homologs, such as Sav1866 and MsbA, are also discussed.

### 2.1. MRP1

MRPs (ABCC) are present in different tissues and are responsible for transporting various types of endo- and xenobiotics [107,108]. They not only function as drug efflux pumps for MDR but also have physiological roles such as detoxification, stress management, inflammation, and substance transportation [109]. Nine members of the ABCC family are known as MRPs. They are classified into long (ABCC1-3, ABCC6, and ABCC10) and short (ABCC4, ABCC5, ABCC11, and ABCC12) families [110].

MRP1 plays a role in protecting the body by pumping out endogenous substances and xenobiotics [111,112,113]. In normal cells, it functions as an efflux pump for the detoxification of xenobiotics generated by phase II enzymes and is present in several important tissues, such as the blood-brain barrier, lung, testis, kidney, intestinal epithelium, peripheral blood mononuclear cells, skeletal and cardiac muscle, and the placenta [114,115,116]. However, overexpression of MRP1 can occur in tumor cells, leading to reduced intracellular concentrations of antineoplastic drugs in cancer cells.

#### 2.1.1. Structure of MRP1

The ABCC1 gene encodes the ATP binding cassette (ABC) transporter known as multidrug resistance protein 1 (MRP1). MRP1, similar to other ABC transporters, has TMDs and NBDs as structural elements. Each TMD has six transmembrane α-helices, and the NBDs are basolaterally localized [117]. MRP1 (1531 residues, 190 kDa [118]) also has a third TMD, called TMD0, with an extra-cytosolic NH2-terminus containing five predicted transmembrane segments [119] (Figure 4). TMD0 is found in the long group of ABCC transporters, including ABCC1-3, ABCC6, and ABCC10 [120]. The entire structure of MRP1, including the three TMDs and two NBDs, forms a stable network to enable domain-domain interaction, which facilitates the binding of substrate drugs in the cytoplasm and expels them to the extracellular milieu via ATP hydrolysis. TMD0 is not essential for the transport function or proper routing of MRP1 to the plasma membrane [57,121]. Lasso motif ($L_{0}$) as a conserved motif in all ABCC class of transporters [116] seems to play a role in the folding of the transporter and has a crucial role in MRP1 regulation [57].

MRP1 has been found to have a single ATPase site that can catalyze a reaction [122,123]. It has two nonequivalent NBDs. Once two ATP molecules connect to NBD’s ATP sites, ATP hydrolysis first takes place in NBD2. Although, glutamate residue in walker B motif is responsible for attracting γ-phosphate of ATP in order to hydrolyze ATP through breaking phosphodiester bond between γ and β-phosphate [86,124], in MRP1, NBD1 has aspartate (D) instead of glutamate (E) residue, which aspartate may not be able to interact effectively with the water because of its short side chain. This distinction is exclusive to NBD1, while glutamate (E) can be found in this position in NBD2, thus, lending it a higher ability to hydrolyze the ATP. Moreover, NBD2 signature motif in MRP1 has a different sequence (LSVGQ) [57,122,125,126]. Additionally, NBD1 in MRP1 has a 13 amino acid deletion, which makes the interaction of this domain (NBD1) to TMD different and weaker compared to the interaction between NBD2 and TMD. The model is called “ball-and-socket”, where the CH of TMD (ball) interacts with the surface of an NBD’s cleft (socket). Therefore, there are fewer interactions between NBD1 and TMD due to the absence of 13 residues in the socket of NBD1 [57,127].

#### 2.1.2. Substrate Binding Site of MRP1

When MRP1 functions as an efflux pump, the NBDs undergo a conformational change that causes a global conformational change, leading to a change in the shape of the substrate-binding pocket [128]. Cryo-EM studies have revealed the structures of MRP1 in two conformations: an apo form and a substrate-binding form [57,129]. These structures can explain the substrate specificity and polyspecificity of MRP1, as the TMDs are hydrophobic, and MRP1 substrates are typically amphipathic organic acids [130,131]. The substrate polyspecificity of MRP1 is achieved through the formation of a single bipartite binding site, which is flexible and predominantly composed of positively charged amino acids (P-pocket). MRP1’s binding site also has a hydrophobic side or H-pocket, which is able to interact with hydrophobic substrates. This binding site allows MRP1 to recognize substrates with various chemical structures, especially organic acids [57,131].

The MRP1 protein provides cellular protection against oxidized heavy metal anions and can also regulate ion channel activity, including an increase in potassium ion channel activity with increased MRP1 expression [132]. MRP1 transports substances that are conjugated to glutathione (GSH), glucuronic acid, or sulfate, such as leukotrienes, alkylating agents, steroids, prostaglandin A2, bile salt derivatives, and folic acid [129,133,134]. Specifically, MRP1 can transport leukotriene C4 (LTC4), a product of arachidonic acid oxidation that is conjugated to antioxidant glutathione (GSH) and can cause bronchoconstriction and pulmonary edema [135]. MRP1 is highly expressed in white blood cells, lungs, and the trachea, where leukotriene synthesis occurs. This indicates that the transport of LTC4 may be an important physiological function of MRP1 [136,137]. In cancer cells, MRP1 can mediate multidrug resistance by promoting the efflux of glutathione-conjugated drugs [131]. MRP1 may also play a role in tumor invasion, metastasis, and disease outcome. MRP1 is involved in the blood-brain barrier and protects intracranial tissues from chemotherapeutic drugs [138]. Additionally, it is involved in mediating inflammation and removing toxic substances from the body [139].

#### 2.1.3. AAM in MRP1

MRP1, present in all human tissues, is responsible for exporting endogenous and exogenous compounds, including arachidonic acid metabolites and drugs such as anticancer agents and antidepressants. Cryo-EM reconstructions of bovine MRP1 have provided insight into how the transporter is able to export a wide range of substrates and how these substrates stimulate ATP hydrolysis [57]. As mentioned above, the substrate-binding site is divided into a hydrophobic region (H-pocket) and a positively charged area (P-pocket). In the presence of ATP, the site changes to a low-affinity state, indicating that substrate release is independent of ATP hydrolysis [140]. The studies have shown that MRP1, in the absence of ATP or any substrate/drug, assumes an inward-facing conformation with the NBDs oriented away from each other and the translocation pathway near the cytoplasm [131]. When the TMDs approach each other, high substrate binding affinity is observed. Upon substrate binding, the NBDs move even closer to form a dimer, resulting in the outward-facing conformation of the ABC transporter [54,131]. Additionally, the extracellular ends of the TMD helices peel outward, causing the residues involved in generating the substrate-binding cavity to be pulled apart. As a result, the transporter’s binding affinity for the substrate decreases significantly, leading to the efflux of the substrate/drug into the extracellular space [141].

The first phase of the transition from the protein’s IF to occluded form is facilitated by the substrate, while the second step is carried out by ATP, which transfers the protein’s occluded to the outward form of MRP1 [57,75]. A recent computational study investigated the transformation of MRP1 from IF to the occluded-facing form through co-transporting GSH and anticancer drugs such as mitoxantrone, epirubicin, and vincristine. GSH is required in most of the transportation by MRP1 [142]. They discovered that two TMHs (TM11 and TM17) of MRP1, GSH, and anticancer medications combine to form a structure known as a sandwich-like structure that causes the movement of transmembrane helices and NBDs; therefore, it causes the formation of the occluded-facing state. The formation of this sandwich-like structure is facilitated by some residues located on TM11, TM17, and TM6 that interact with antitumors and GSH in substrate binding sites of the MRP1. Trp1246 is the residue on TM17 in H-pocket, forming a bond with anticancer drugs; however, Lys332 of TM6 interacts with GSH in the P-pocket. A linker also forms between TM11 and TM17 through Phe594 of TM11. The simulations of the systems with anticancer drugs have been done in the absence and presence of GSH. The results indicated that MRP1 undergoes large conformational changes in the presence of both GSH and anticancer drugs. For instance, the distance between NBDs decreased, while ATP-binding sites between NBD1 and NBD2 were formed. However, this is not observed in the absence of even one of them. Some experimental evidence shows GSH is necessary to transport some of the drugs by MRP1 [132,143]. Zhao et al. (2020) also proposed unidirectional movement of NBD2 into NBD1 and believed that NBDs in MRP1 do not move toward each other like tweezers [93].

Johnson et al. (2018) investigated the role of ATP binding and hydrolysis in the release of substrates from the MRP1 by employing a combination of biochemical assays and molecular modeling. As a result, they found that releasing the substrate does not need ATP to be hydrolyzed; however, the hydrolysis of ATP is necessary to reuse the transporter for the next cycle [54].

A research study by Degorter et al. (2008) focused on the transport of drugs using the homology model of MRP1 with OF crystal structure of Sav1866 and conducted three different mutations on Y324 residue (Y324W, Y342F, Y324A), which is located in transmembrane helix number 6 of TMD1 (TM11). The results showed that systems with mutations to Alanine or tryptophan behaved similarly to WT systems, whereas a mutation to phenylalanine (Y342F) enhanced MRP1’s activity to transfer the medicines [144]. Based on these results, Amram and colleagues started working on isolated TMDs [145]. They repeated the previous work with isolated TMDs in the POPC membrane. Once again, the Y342F system showed a higher clearance rate by remaining open among all the systems. According to research conducted by Weigl et al. (2018), the residue Phe583, located in TMD1 between TMH10 and TMH11, is critical for its transport function. The F583A mutation in MRP1 inhibits ADP release, which traps the transporter in a substrate-releasing state and reduces its ligand affinity [146].

### 2.2. P-glycoprotein

P-glycoprotein (Pgp), which is also known as ABCB1 [43], is encoded by the MDR1 gene [147]. It is a transmembrane protein expressed in many tissues, including the cells lining the intestine, liver, and kidney [148,149]. It is a member of the ATP-binding cassette (ABC) transporter superfamily [150,151,152] and plays a crucial role in protecting the body from toxic substances by exporting them out of cells [153]. However, Pgp also causes multidrug resistance in cancer cells by pumping out anti-cancer drugs [154] including anthracyclines, taxanes, and vinca alkaloids [155,156], thereby reducing the effectiveness of these drugs, which makes it an important target for drug development [157]. Pgp functions as an efflux transporter by recognizing and binding a wide range of structurally diverse drugs and pumping them out of the cell [158,159]. The transport of drugs by Pgp is driven by the hydrolysis of ATP [62,160,161], which induces a conformational change in the protein and facilitates the movement of drugs across the membrane [160,162,163,164]. Dysregulation of Pgp has been implicated in various diseases, including cancer, Alzheimer’s disease, and epilepsy [165,166]. In cancer, Pgp is known to play a key role in the development of drug resistance in cancer cells [155]. By pumping out chemotherapeutic drugs, Pgp makes tumors resistant to chemotherapy, thereby reducing their effectiveness in killing cancer cells [156]. Therefore, overexpression of Pgp in cancer cells can lead to multidrug resistance [165,167], which makes it difficult to treat cancer with a variety of different drugs [167]. It is also reported that in the central nervous system, Pgp plays a role in regulating the entry of drugs and toxins into the brain [168,169], and its dysregulation has been implicated in neurological disorders [170].

#### 2.2.1. Structure of P-glycoprotein

Pgp is a 1280-residue [171], 170 kDa membrane protein [147,172,173,174] that exists as a heterodimer with pseudosymmetrical characteristics [175], and consists of two TMDs and two NBDs, one in each monomer [176]. Within Pgp, there exist two operational ATPase sites. Each site consists of the Walker A and Walker B motifs from one NBD, as well as the LSGGQ motif from the other NBD. At both of these sites, a glutamate residue, which is significantly conserved, functions as the catalytic base for ATP hydrolysis. If either of these catalytic glutamates undergoes mutation, the ATPase activity is severely diminished. Simultaneous mutations at both positions result in the trapping of Pgp in an ATP-occluded form, potentially with a closed-NBD dimer [62,177]. The NBDs possess multiple conserved sequences that play a crucial role in nucleotide binding and hydrolysis. In an NBD dimer, each of the two nucleotide-binding sites is formed by combining the Walker A motif, Walker B motif, A loop, H loop, and Q loop of one NBD with the D loop and signature motif of the other NBD. The Walker A motif is responsible for nucleotide phosphate binding, while the Walker B motif, along with the Q loop, coordinates Mg^2+^ and water at the catalytic site. The A loop contains an aromatic residue that interacts with the adenine ring of ATP, and the signature motif, D loop, and H loop contribute to the coordination of the ATP γ phosphate [128].

The general structure of this protein is TMD1-NBD1-TMD2-NBD2, with both the N-terminal and C-terminal of the protein located in the cytosol. In the inward-facing and outward-facing structures of human Pgp, the estimated distances between the C-terminal of NBD1 and NBD2 are approximately 30 Å and 11 Å, respectively [178]. The TMDs which form the substrate translocation pathway [179], are composed of six α-helices each and form a large central cavity that serves as the drug-binding site [172,180]. The NBDs contain the ATP-binding sites and are responsible for the energy-dependent transport of drugs across the membrane [181].

#### 2.2.2. Substrate Binding Site of P-glycoprotein

Pgp substrates typically have weak amphipathic properties, are relatively lipophilic, and contain aromatic rings. They are also generally positively charged at normal pH levels [128,182,183]. The substrate binding pocket of Pgp is a large, highly complex, and polyspecific structure that plays a crucial role in substrate recognition and transport [174]. One of the most important features of the substrate binding pocket of Pgp is its ability to undergo conformational changes [184]. The pocket can adopt different conformations depending on the size and shape of the substrate. This flexibility is essential for accommodating structurally diverse substrates and maintaining a high affinity for binding them [184]. The substrate binding pocket of Pgp contains a network of hydrophobic and aromatic residues [185]. The hydrophobic residues include phenylalanine, leucine, and tryptophan, while the aromatic residues include tyrosine and tryptophan. These residues interact with the hydrophobic regions of the substrate and help to stabilize its binding within the pocket [172]. The degree of hydrophobicity observed in the drug-binding pocket of mouse Pgp is significantly greater than that observed in Sav1866 [186]. Therapeutic drugs, such as antiarrhythmic agents, calcium channel blockers, analgesics, chemotherapeutic drugs, HIV-protease inhibitors, antihistamines, antibiotics, and immunosuppressive agents, are among the Pgp substrates [128,187,188,189].

#### 2.2.3. AAM in P-glycoprotein

The association of Pgp with multidrug resistance in cancer cells and the treatment failure of various drugs has led to an increased interest in understanding its mechanism of action [155,156]. The mechanism by which Pgp transports drugs across the cell membrane is called the alternating access mechanism [190,191]. The alternating access mechanism is a widely accepted model for the function of Pgp [174,190]. It suggests that the protein has two major conformations, each with a different orientation towards the inside or outside of the cell [40]. In the inward-facing conformation, the drug-binding pocket is accessible from the cytoplasmic side of the membrane. The drug molecule binds to the pocket and triggers a conformational change that moves the protein toward the outward-facing conformation. In the outward-facing conformation, the drug-binding pocket is accessible from the extracellular side of the membrane [172]. The switching between these two conformations is controlled by ATP hydrolysis, which provides the energy required to move the protein between the two conformations [62,160]. When ATP binds to Pgp, it causes a conformational change that results in the release of the drug molecule outside of the cell [171,192,193]. After ATP hydrolysis, Pgp undergoes a conformational reset and can bind to another drug molecule [162,163,164].

Electron microscopy and crystallographic studies have revealed the structure of the protein in both the inward- and outward-facing conformations [62,194,195,196], while Molecular dynamics (MD) simulations have been used to study the alternating access mechanisms [43,196,197]. MD is a powerful computational technique used to study the dynamics of biological macromolecules such as proteins [198,199], nucleic acids, and lipids [179,200,201]. It is an essential tool for studying the mechanisms of membrane transport proteins [106,184,202,203], including the alternating access mechanism in Pgp [43,196,197].

A recent study by Li et al. (2022) employed the two-state anisotropic network model to investigate the allosteric pathway of human Pgp in the substrate transport process, specifically from the inward-facing (IF) to the outward-facing (OF) state [204]. The authors found that the allosteric transitions proceed in a coupled manner and ultimately lead to conformational changes in the NBDs, resulting in the TMDs moving to the OF state via the allosteric propagation of intracellular helices IH1 and IH2. They indicate that the allosteric transition begins with the large-scale closing motion of the NBDs, accompanied by a significant twisting movement between them. This twisting motion becomes more pronounced near the transition state and is transmitted to TMDs via IH1 and IH2 intracellular helices. In conclusion, they reveal that this allosteric pathway is energetically favorable compared to alternative methods [204].

Another study by Vahedi et al. (2017) used computational analysis to investigate the significance of tyrosine residues in regulating Pgp-ATP hydrolysis via hydrogen bond formations with high-affinity modulators. The authors found that the mutation of 15 conserved residues in the drug-binding pocket of human Pgp to tyrosine resulted in no major effect on the total or cell surface expression of the mutant, indicating that tyrosine is critical in the drug-binding pocket and transport function of Pgp [205]. Verhalen et al. (2017) utilized double electron-electron resonance (DEER) and molecular dynamics (MD) simulations to investigate the ATP- and substrate-coupled conformational cycle of the mammalian ABC efflux transporter Pgp. They did this by introducing pairs of spin labels at selected residues to monitor the transition from an inward-facing (IF) to an outward-facing (OF) state. Their findings showed a two-stroke cycle in which ATP energy is harnessed in the NBDs, leading to conformational changes that reconfigure the TMD [43].

In addition to these studies, several other investigations have contributed to our understanding of the alternating access mechanism in Pgp. For example, a study by Futamata et al. (2020) used a FRET sensor to monitor the roles of ATP binding and ATP hydrolysis in the conformational changes of Pgp under a physiological membrane environment. The authors found that the ATP binding induces the conformational change to the outward-facing state, while ATP hydrolysis and subsequent release of γ-phosphate from both NBDs allow the outward-facing state to return to the original inward-facing state [178].

In determining the Molecular structure of human P-glycoprotein in the ATP-bound, outward-facing conformation, Kim et al. (2018) explained that the transition from inward-facing to outward-facing conformation involves a global movement of both halves of the molecule and extensive local rearrangements of the TM helices. They illustrated that the two “crossing” helices in each TMD (TM4 and 5 in TMD1, and TM10 (TM4) and TM11 (TM5) in TMD2) pivot inward, bringing the NBDs closer to each other. Additionally, the extracellular regions of TM7 (TM1) and 8 (TM2) move away from TM9 (TM3) to 12 (TM6), resulting in an outward-facing configuration [62] (Figure 5). The TM helices in TMD2 can be numbered from 1 to 6 or 7 to 12.

Loo et al. (2013) predict that a salt bridge formation between Glu256–Arg276 could enhance the folding of human Pgp by facilitating the packing of the TM segments. They explain the possibility as resulting from the linking of TM4 and TM5 segments by intracellular loop 2 (ICL2). The proposed function of the ICLs is to act as “ball-and-socket” joints that connect the TMDs to the NBDs and serve as transmission interfaces. ICL1 (TMD1-NBD1), ICL2 (TMD1−NBD2), ICL3 (TMD2−NBD2), and ICL4 (TMD1−NBD2) facilitate connections between domains [206]. In the crystal structure of *C. elegans* Pgp, ICL2 was predicted to be the most crucial link between the TMDs and NBDs in human Pgp [207], as 14 amino acids in this loop were projected to connect TMD1 to NBD2 via salt bridges, hydrogen bonds, or Van der Waals interactions. The structure of ICL2 is vital for Pgp biosynthesis since point mutations in this region hinder maturation [206].

Researchers in an experimental study investigated the role of the coupling helices (CH1 and CH2) in Pgp by conducting point mutations on these specific regions of the protein. They discovered that the mutations on CH2 have negative effects on Pgp activity; however, these specific mutations do not inhibit the function of Pgp when they are on CH1 [208]. According to the results of dynamic simulation analysis of human Pgp for generating the required conformational changes of this transporter, CH2 interactions with NBDs are higher and play a more important role compared to CH1 [171].

Studies have demonstrated that lipid molecules (notably cholesterol) have a significant impact on regulating the structure and function of membrane proteins such as GPCRs [201] and membrane transporters, such as Pgp [209,210]. Pgp has been shown to have the ability to act as a lipid flippase, which means it can move cholesterol and phospholipids between the inner and outer layers of the cell membrane, redistributing and shuttling them [209]. By employing MD simulations, Thangapandian et al. (2020) investigated the role of cholesterol binding and translocation in Pgp [209]. From their simulations, they observed that the high-density binding region for cholesterol is located on the first transmembrane domain bundle of Pgp, specifically at the interface between TM1 and TM2, resulting in asymmetric cholesterol accumulation [209]. The study found that cholesterol binds more frequently to Pgp through its rough β face formed by the two protruding methyl groups, whereas the opposite smooth α face prefers packing alongside the membrane lipids. Additionally, the simulations captured one full and two partial cholesterol flipping events between the two leaflets of the bilayer mediated by the surface of Pgp. These events were observed in a region formed by helices TM1, TM2, and TM11, which featured two full and two partial CRAC/CARC motifs. During these events, Tyr49 and Tyr126 were identified as key residues that interacted with cholesterol. Experimentally, Clouser et al. (2020) aimed to investigate how cholesterol in a membrane impacts the conformational behavior of Pgp in lipid nanodiscs. Using hydrogen/deuterium exchange mass spectrometry, they found that cholesterol in the membrane causes asymmetric, long-range modifications in the distribution and exchange kinetics of conformations of the nucleotide-binding domains. They also showed that cholesterol presence enhances ATP hydrolysis and modifies lipid order and fluidity. These modifications link the effects of lipid composition on activity with specific changes in the Pgp conformational landscape [210].

These recent studies provide further insights into the alternating access mechanism in Pgp and identify the major domain sites, specifically the NBD and TMD, involved in the conformational changes that occur during substrate transport. Further research in this area will help to deepen our understanding of the mechanism of Pgp and may provide insights into new approaches for drug development.

### 2.3. Sav1866

Multidrug resistance (MDR) is a major obstacle in the treatment of bacterial infections, where pathogens become resistant to multiple antibiotics [211,212,213]. One of the key mechanisms of MDR is the over-expression of a protein called Sav1866 [214]. Sav1866 acts as a transporter, exporting a wide range of substrates, including drugs, chemotherapeutic agents, peptides, and lipids, out of bacterial cells and reducing their efficacy [215]. This qualifies it as an attractive target for understanding the mechanisms of drug resistance and developing strategies to overcome multidrug resistance (MDR) in bacteria and other microorganisms [165].

#### 2.3.1. Structure of Sav1866

Sav1866 are found in various bacteria and archaea [216], including *Staphylococcus aureus* and *Mycobacterium tuberculosis* [217]. Sav1866 shares structural and functional similarities with eukaryotic ABC transporters, such as Pgp, and can serve as a model for understanding the mechanisms of drug resistance in both prokaryotic and eukaryotic organisms [218]. The structure of Sav1866 exists as a homodimer [67,179], consisting of two TMDs and two NBDs that are connected by a linker region [66,67]. The TMDs form a substrate translocation pathway, while the NBDs provide energy for substrate transport [179,219,220]. Following a recent classification of ABC transporters based on their distinct TMD folds, Sav1866 (Figure 6) belongs to the group IV classification based on their structural homology in the TMDs [42].

#### 2.3.2. AAM in Sav1866

The alternating access mechanism allows for the selective transport of substrates while maintaining the integrity of the cell membrane [221]. The alternating access mechanism in Sav1866 involves two distinct conformations of the protein, referred to as the inward-facing conformation and the outward-facing conformation [165,179,186,222,223], with a third intermediate phase described as occluded conformation [179]. In the inward-facing conformation, the TMDs are closed, and the substrate-binding site is accessible from the intracellular side of the membrane. When ATP is bound and hydrolyzed by the NBDs, the protein undergoes a conformational change to the outward-facing conformation [224,225]. In this conformation, the TMDs open to the extracellular side of the membrane, and the substrate is released into the extracellular environment [216,219,223,226]. The opening of the TMDs involves the swapping of TMs 1 and 2 in the outward-facing dimer and TM 4 and 5 in the inward-facing dimer [39,103]. Therefore, the alternating access mechanism in Sav1866 alternates between an apo inward to a nucleotide-bound outward-facing conformation to transport their substrates across cellular membrane [179].

The conformational changes in Sav1866 that underlie the alternating access mechanism have been studied using a variety of techniques, including X-ray crystallography, electron microscopy, and molecular dynamics simulations [66,165,179,226]. These studies have provided valuable insights into the structural changes that occur during substrate transport and have helped to explain the alternating access mechanism in Sav1866. For example, the crystal structure of Sav1866 in the outward-facing state was determined by Dawson and Locher (2006) using X-ray crystallography [66]. It was revealed that Sav1866’s structure comprises of TMDs that create a path for substrate translocation and two NBDs that supply the necessary energy for substrate transport. The substrate translocation pathway is closed in the outward-facing state, preventing the substrate from entering or leaving the binding site.

In the inward-facing state, the substrate translocation pathway opens, allowing the substrate to enter or leave the binding site. The switch between the outward-facing and inward-facing states is mediated by the movement of the TMDs and the coupling of this movement to the NBDs. The binding of ATP to the NBDs induces a conformational change in the TMDs, which results in the opening or closing of the substrate translocation pathway [66,67].

Molecular dynamics simulations have also been extensively used to study the alternating access mechanism in Sav1866 [165,179,224]. They have revealed that the protein undergoes a series of conformational changes involving the movement of helices and loops in the TMDs, as well as the repositioning of the NBDs, during substrate transport [179,219]. St-Pierre et al. (2012) employed MD simulations to investigate the stability of Sav1866 and study the ability of the protein to undergo conformational changes at physiological temperatures. They found that the protein transmembrane domain is not easily disturbed by large-scale motions of the NBDs [165].

Other studies have focused on the role and dependency of various lipids in the AAM of Sav1866 [179,227]. The composition of lipids has been known to alter the structural and functional dynamics of membrane proteins and membrane transporters [179]. For example, Immadiesity et al. (2019) used microsecond-level all-atom MD simulations to study the lipid-dependent characteristic of the alternating access mechanism in Sav1866. Their study focused on the conformational changes Sav1866 undergoes from an OF-IF transition in six different lipid environments. They show that the protein undergoes a large-scale conformational transition in the PE lipid bilayers, particularly in the POPE lipids, resulting in an IF-occluded conformation on the periplasmic side; however, this is not observed when the transporter is embedded in any of the PC environment [179]. The study identified a salt bridge (R81-D145) (Figure 7A) between two helices (TM2 and TM3) in the same monomer near a hinge region in one of the helices in POPE simulations, which was not observed in the crystal structure. The formation of this salt bridge was accompanied by the weakening of another salt bridge between TM6 and TM3 of the same monomer was present in the crystal structure: R296-D145 (Figure 7A), leading to the formation of the hinge region in TM3. The events were observed in one monomer in POPE simulations and were likely to promote periplasmic closure. The salt bridge brings the two bundles closer to each other while the hinge region allows for easier bending of one of the helices, which could ultimately lead to a closure of the periplasmic gate [179].

Aittoniemi et al. (2010) focused on the role of specific residues in the AAM of Sav1866 and showed that MgATP-binding residues and a network of charged residues at the dimer interface form a sequence of putative molecular switches that allows ATP hydrolysis only at one NBS [228]. The simulations found an asymmetry in the dynamic behavior of two identical Sav1866 monomers, with certain interactions only occurring in one monomer. These interactions involved the formation of a salt bridge (D423-K483) (Figure 7B), which has also been reported to occur in only one monomer of the homologous MgATP-bound NBD homodimer HlyB [229]. Notably, while the salt bridge was formed by residues within the same NBD monomer in HlyB, it is formed between the two NBDs in Sav1866. The formation of this salt bridge may be structurally coupled to conformational changes during ATP hydrolysis. The interactions could be dependent on the association of a Q-loop Gln residue (Q422) with MgATP, and when unbound from K483, D423 can interact with the coupling helix on the cytosolic loop (CL) of the opposite monomer, tightening NBD-TMD interactions. The formation of these interactions is preceded by the breaking of the D423 charge pairs with R474 of the short sequence motif in the NBD-TMD interface named X-loop [228]. Multiple computational studies on Sav1866 indicated that CHs forming interaction between TMDs and NBDs play a significant role in conformational changes of this transporter [224,228].

Overall, the AAM of Sav1866 is of significant interest because it is a model system for understanding the mechanisms of substrate transport by ABC transporters. The structural and functional studies of Sav1866 have provided valuable insights into the design of new antibiotics that can target the transporter specifically [217,230].

### 2.4. MsbA

In Gram-negative bacteria such as *Escherichia coli*, *Acinetobacter baumannii*, and *Salmonella typhimurium*, MsbA plays a crucial role as an ATP-binding cassette transporter [231]. MsbA, as a multidrug-resistant transporter (MDR), is crucial for the movement of drugs; it is also able to transfer other small molecules such as phospholipids and dyes [60,232]. Translocation of core-LPS is one of the functions mediated by MsbA [233]. Gram-negative bacteria are resistant to many antibiotics due to their outer cell membranes being permeabilized by lipopolysaccharide (LPS) [234], which consists of a hydrophobic lipid A anchor, a non-repeating core oligosaccharide, and a distal oligosaccharide [235]. MsbA transports lipid A and lipopolysaccharide across the inner membrane from the cytoplasmic to the periplasmic leaflet [236]. The core-LPS molecule is synthesized in the inner leaflet of the bacterial membrane, enters the MsbA transporter, and is flipped to the periplasm for the further process [69].

#### 2.4.1. Structure of MsbA

MsbA (130 kDa, 584 residues [233,237]) is a homodimer that consists of two TMDs, each of which has six transmembrane (TM) helices and two cytosolic NBDs [69]. This protein is classified in group IV based on TMD fold arrangement (Figure 1). The TMDs of MsbA are responsible for accommodating various substrates and facilitating their translocation. Although the NBDs remain highly conserved, the TMDs vary in structure and sequence. MsbA’s TMDs are made up of 12 transmembrane helices (TMHs), which form two wings that change their organization upon switching between the inward-facing (IF) and outward-facing (OF) conformations [71,238,239]. The core-LPS binding site, which accommodates the phosphorylated glucosamines(P-GlcN), acyl chains, and core sugars (core oligosaccharides) of the substrate, is formed by the TMHs and is accessible via a lateral portal formed by TMH4/6 of each monomer [68,69].

#### 2.4.2. Substrate Binding Site of MsbA

The interior space of MsbA consists of a hydrophilic cavity and a hydrophobic pocket, which are exposed to the cytoplasmic and periplasmic sides, respectively. The P-GlcN and acyl chains of LPS interact with the hydrophobic inner cavity of MsbA; however, the inner core is located in a hydrophilic cavity. The interaction of the outer core is unclear [71,240].

According to a cryo-EM study [71], glucosamines have hydrophilic interactions with amino acids (Arg78, Arg148, Gln256, and Arg296 and Lys299) located at TM helices (TM2, 3, 5 and 6). They also mentioned that mutations in three conserved residues (Arg78, Arg148, and Lys299) to alanine affect lipopolysaccharide (LPS) binding. Discussions about MsbA-mediated LPS transport mechanisms are still ongoing. A study by Mi et al. (2017) proposed a model for MsbA flippase named “trap and flip” mediating LPS transport. The IF state of MsbA occurs in the absence of nucleotide or the presence of ADP in this model. The LPS substrate enters the binding site located in TMD and rearranges MsbA to allow ATP to bind to NBDs. The acyl chains of LPS are able to enter the periplasmic leaflet and trigger core oligosaccharides to move upward. The MsbA conformational changes and the translocation of LPS result in ATP hydrolysis and resume the IF state of MsbA [71].

#### 2.4.3. AAM in MsbA

Across-membrane substrate transfer by MsbA is known to occur via an alternating access mechanism. The inward conformation undergoes a transition to the outward conformation upon binding of two ATPs and the substrate. A computational study by Moradi et al. (2013) sheds light on the study of large-scale conformational changes in ABC transporters such as MsbA using nonequilibrium MD simulations. This study highlights the critical role of TMD-NBD interactions in alternating access mechanisms that govern the function of MsbA. They used multiple collective variables to induce the orientation of NBDs and TMDs and induce the transition OF to IF state [241]. A combination of nonequilibrium simulations and free energy calculations for investigating large-scale conformational changes in proteins is proposed in another computational study by the same authors. Since the IF to OF transition occurs over extremely long time scales, MD simulations cannot be used to directly investigate this phenomenon. As MsbA crystal structure exists in three states: OF, IF-open, and IF-closed (Figure 8), it can be used to investigate the transition by employing collective variable (CVs) and biasing protocols to bias the system towards the specific states. They mentioned an asymmetrical arrangement of MsbA’s TMDs based on a significant salt bridge between residues D252 (TM5) and K299 (TM6) in the IF state. They could find this salt bridge once in the IF form of MsbA instead of twice or none [242]. This finding agrees with experimental evidence indicating the asymmetrical behavior of TMDs in MsbA [243].

In addition to accommodating the substrate, the TMDs must communicate with the NBDs for substrate release [244]. This communication is facilitated by intracellular loops called coupling helices (CH1/2), located at grooves on the NBD surface [245,246,247]. A study has been done on the MsbA transporter using Molecular dynamics simulations and enzyme assays to investigate the role of the coupling helices (CHs) at the interface between the NBD and the TMD. They used MsbA and mutated either CH1 or CH2 residues to alanine (A). The experimental and computational results indicated that ATP affinity is decreased for both systems, including mutations on either CHs. However, the ATPase activity of only mutated CH2 is reduced as conserved His357 of H-loop in ATP binding pocket (ABP) had fewer interactions with the γ-phosphate of ATP. Hence, both CHs contribute to ATP binding, but CH2 plays more significant roles in ATPase function [247].

TMH3/4 of each TMD form a tetrahelix bundle at the intracellular extensions of the coupling helix (CH), which is essential for ATPase activity and conformational changes related to ATP binding in the TMD region [248].

In MsbA, both of NBDs have all conserved motifs (Walker A/B, Signature Motif, A/D/Q-loop, H-Switch) as their ATP-binding pocket (ABP), and they can be dimerized by turning from the top down (head-to-tail manner) and sandwiching two ATP molecules between them [70,247]. Hydrolysis of ATP takes place in MsbA via both NBDs, as both contain all required motifs; however, whether this happens simultaneously is unclear.

NBD of MsbA in *Escherichia coli* was the subject of a study that integrated site-directed mutagenesis and biochemical assays. The results of this study indicate that the mutations in particular amino acid residues of MsbA’s nucleotide-binding domains (e.g., Leucine 504 of Walker B motif) significantly alter the protein’s ability to bind and hydrolyze ATP, as well as affect the rearrangement of the transporter [249].

Under circumstances of ATP depletion, NBDs in MsbA can also perform the reverse adenylate kinase reaction (AK reaction), which involves the conversion of ADP to ATP and AMP (2ADP ↔ ATP + AMP) [250,251,252]. There is an experimental study on MsbA proposing a novel model called “coupled ATPase-adenylate kinase mechanism” [253]. This study suggests that ATP hydrolysis in some ABC transporters, such as MsbA, is coupled with the activity of adenylate kinase. The AK reaction employs the conserved NBD motifs that are also required for ATP hydrolysis. Furthermore, solid-state NMR demonstrates that the Q-loop is involved in transient nucleotide binding during the AK reaction [253,254].

## 3. Conclusions

In this review paper, we have described ATP binding cassette transporter (ABC) classifications and focused on the specific type of ABC transporters named multidrug resistance (MDR) proteins, including mammalian Pgp and MRP1, as well as bacterial homologs of Pgp (Sav1866 and MsbA). We mentioned that the above transporters belong to type IV transporters based on their TMD fold arrangement. The functional core of this class of proteins has two TMDs (each one has six TM helices (TMHs)), and two NBDs, although MRP1 has an extra TMDs named TMD0 (including five TMHs), and its function for the transportation of substrates is not clear yet. In the IF state (Figure 5), each wing of these proteins consists of six helices, four of which come from TMD1 (H1, H2, H3, H6) and two of which come from TMD2 (H4 and H5), whereas in the OF state, the arrangement is changed, with TMHs from domain one including TMH1,2 plus TMH from domain two composed of TMH3, 4, 5, 6. However, the structure of NBDs in prokaryotic ABC proteins such as Sav1866, MsbA, and mammalian Pgp are identical, while NBDs in MRP1 are distinct. The presence of two ATP molecules is required to form an interface between two binding sites of NBD domains and in all of these transporters. ATP hydrolysis takes place following substrate transport and is necessary for reusing the transporters for the next cycle of substrate transportation. NBD2 is the only NBD domain in MRP1 that is able to hydrolyze the ATP molecule; however, both NBDs of Pgp, Sav1866, and MsbA are able to carry out this reaction. Moreover, NBD dimer dissociation can be accomplished with only one ATP hydrolysis. For the full transporters to properly couple dynamically, the coupling helices (CHs) must play a crucial role in the coordinated motion of complete transporters. These findings suggest that CHs mediate interactions at interfaces between NBDs and TMDs, facilitating the propagation of conformational changes from NBDs to TMDs. Furthermore, MsbA is the only one of these four transporters to exhibit the adenylate kinase reaction (AK reaction) associated with NBD.

Understanding the involvement of certain protein domains in inducing conformational changes, such as transmembrane helices (TMHs), is an important area of research with considerable prospects for the development of more effective therapeutic approaches. For instance, mammalian Pgp and MRP1 are well-known contributors to multidrug resistance in cancer chemotherapy. Knowing the structural features and functional mechanisms of these transporters can help develop targeted therapies to overcome drug resistance. We can improve patient outcomes and enhance chemotherapy effectiveness by inhibiting these transporters. Additionally, the structural and mechanistic similarities of Sav1866 and MsbA proteins to human Pgp and MRP1 provide valuable information and make them extremely helpful. These bacterial transporters play a vital role in bacterial drug resistance, and understanding their structure-function relationships can inform the development of novel strategies to tackle multidrug-resistant bacterial infections. Multiple conformations of these transporters exist throughout their transport cycle, and by focusing on particular conformations, researchers can investigate various functional aspects and create a variety of treatment approaches. Namely, inhibiting substrate binding can prevent the efflux of drugs and enhance their efficacy. Inhibiting ATP hydrolysis, on the other hand, can disrupt the energy supply necessary for transporter function. Developing therapies to inhibit the dissociation of NBDs can also impact the transport cycle and disrupt the drug resistance mechanisms.

We expect that the information provided in this review will provide insight into all aspects of mentioned proteins. Although much has been learned about these transporters, there is still much more to discover. More research into individual transmembrane helices (TMHs) and their communication with the NBDs and substrates is needed to better understand the role of specific protein domains in triggering conformational changes.

## Figures and Tables

**Figure 1 membranes-13-00568-f001:**
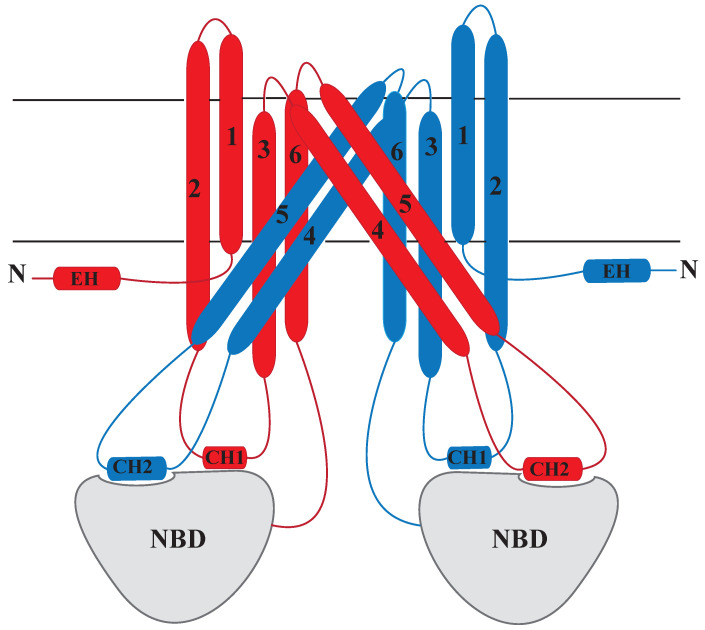
Schematic representation of type IV ABC transporters which are classified based on TMD arrangement. Each TMD (blue and red color) is composed of six transmembrane helices (1–6). CH and EH indicate coupling helix and elbow helix, respectively. In this transporter type, the elbow helix (EH), located at the N-terminus of each transporter half, plays a crucial role in stabilizing the transporter by positioning itself on the membrane-facing surface of the NBD. Each TMD of type IV transporters comprises a conserved core consisting of six transmembrane helices. This structure has two coupling helices (CH1, CH2) that interact with the NBDs.

**Figure 2 membranes-13-00568-f002:**
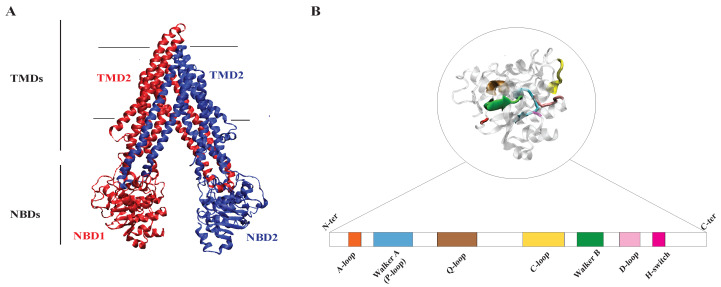
(**A**) The crystal structure of an ABC transporter including TMDs (TMD1 and TMD2) and NBDs (NBD1 and NBD2)(PDB ID:7OTI). (**B**) Schematic representation of NBD, which shows the positions of sequence motifs within NBDs.

**Figure 3 membranes-13-00568-f003:**
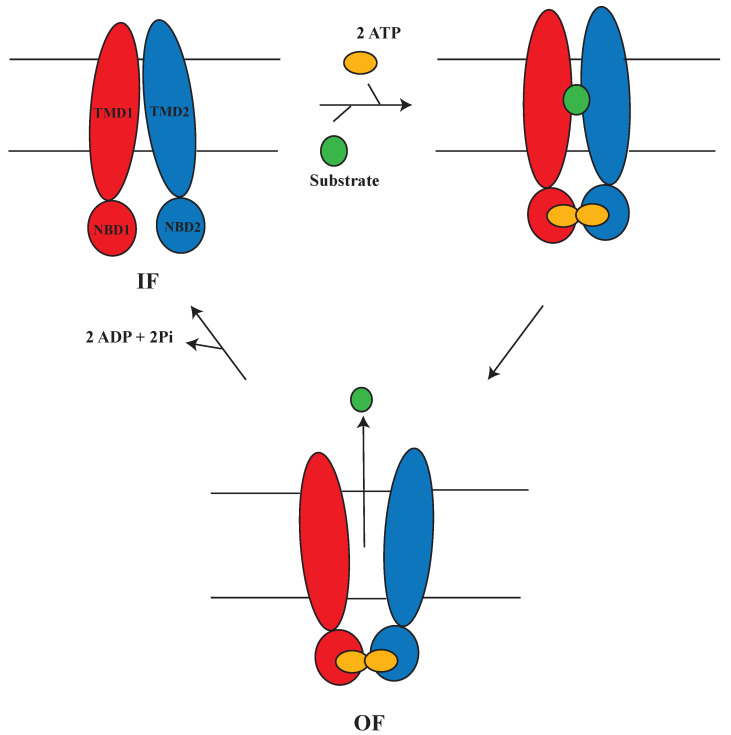
Schematic representation of the alternating access mechanism in ATP-binding (ABC) transporters. The substrate (green circle) initiates the transport cycle by binding to a substrate binding site formed by the TMDs. The NBDs then undergo a conformational change that permits ATP binding (orange ellipse) and the formation of a closed NBD dimer. A significant conformational change in the TMDs is triggered by the closed NBD dimer, and TMDs open toward the outside, allowing substrate translocation to begin. Dissociation of NBD dimer is triggered by ATP hydrolysis, and the transporter returns to an inward-facing conformation state by releasing phosphate and ADP.

**Figure 4 membranes-13-00568-f004:**
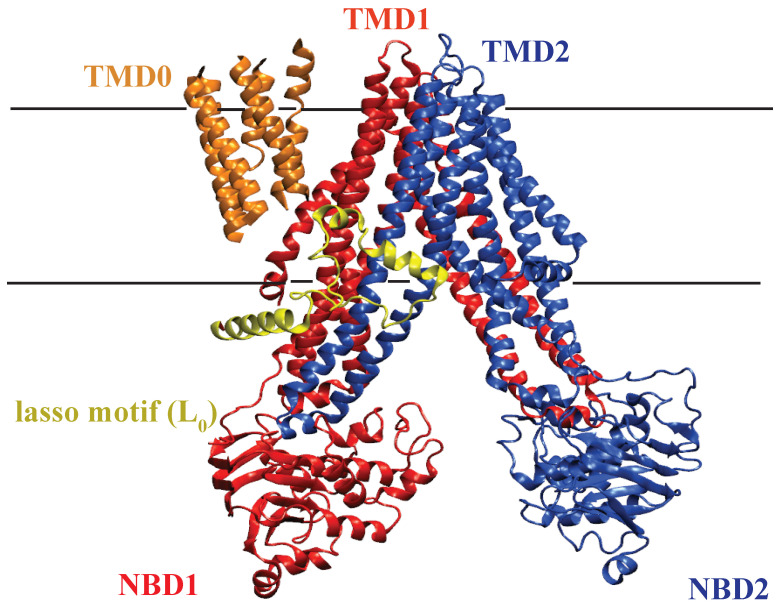
Crystal structure of the MRP1 (PDB:5UJA) including three transmembrane domains (TMD0, TMD1, TMD2), two cytosolic domains (NBD1 and NBD2), and Lasso motif ($L_{0}$) connecting TMD0 to the rest of the protein.

**Figure 5 membranes-13-00568-f005:**
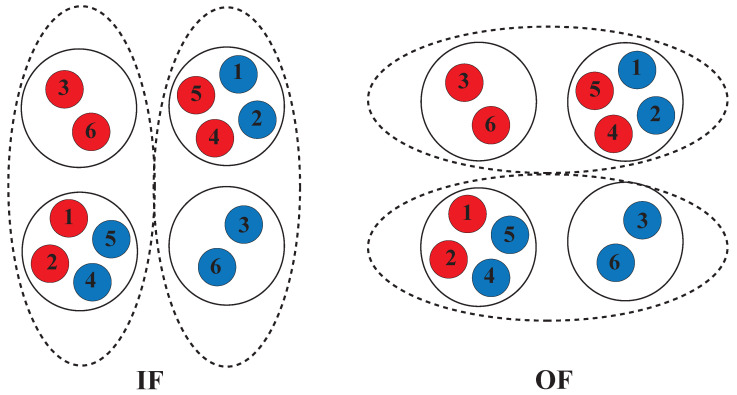
Schematic representation of transmembrane helices (TMs) arranged inward-facing (IF) and outward-facing (OF) orientation. Each TM helix is represented by a numbered circle in blue and red, indicating which subunit they belong to.

**Figure 6 membranes-13-00568-f006:**
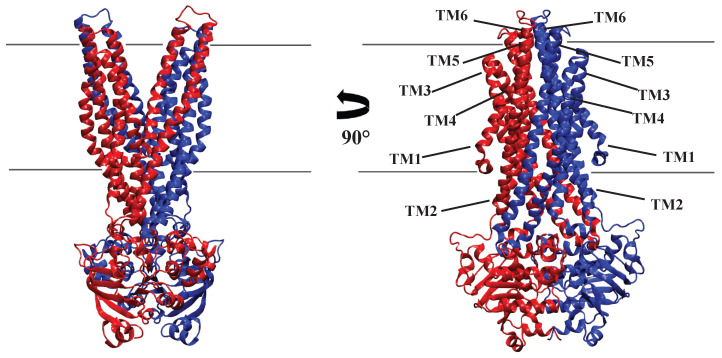
The cartoon representation of Sav1866 (PDB:2HYD) in OF state, showing both monomers in this structure with different colors. Each monomer has six transmembrane helices (TM1–TM6).

**Figure 7 membranes-13-00568-f007:**
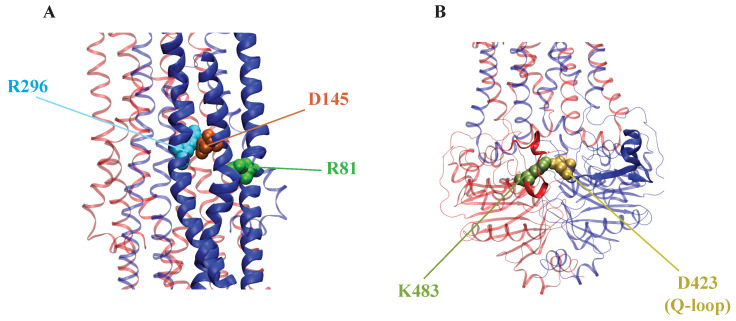
The cartoon representation of Sav1866 (PDB ID: 2HYD) shows the salt bridge interactions in different parts of the protein. Different colors indicate the different amino acids involved in the salt bridge interactions. (**A**) The salt bridge interaction between R81 and D145 of two helices (TM2 and TM3) in the same monomer. This figure also shows salt bridge interaction between R296 and D145 (TM6 and TM3 of the same monomer). (**B**) The salt bridge interaction between D423 and K483 is located in the Q-loop and region beside the signature sequence, respectively.

**Figure 8 membranes-13-00568-f008:**
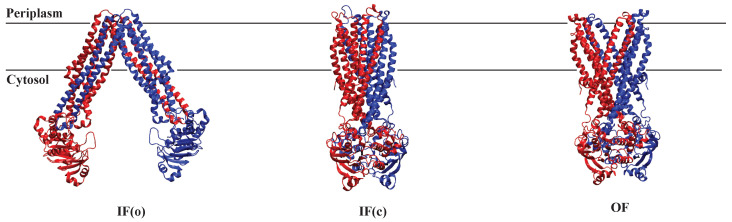
Cartoon representation of MsbA crystal structures in inward-facing open (IF(o)), closed (IF(c)), and outward-facing (OF) conformations. PDB IDs used to generate these structures are 8DMO, 7BCW, and 8DMM, respectively.

**Table 1 membranes-13-00568-t001:** Available PDB files for MRP1, Pgp, Sav1866, and MsbA transporter proteins.

Name	PDB ID *	Resolution	Method **
MRP1	6BHU [54]	3.14 Å	Cryo-EM
	6UY0 [55]	3.23 Å	Cryo-EM
	8F4B [56]	3.27 Å	Cryo-EM
	5UJA [57]	3.34 Å	Cryo-EM
	5UJ9 [57]	3.49 Å	Cryo-EM
	7M68 [58]	4.04 Å	Cryo-EM
Pgp	5KO2 [59]	3.30 Å	X-ray
	5KPD [60]	3.35 Å	X-ray
	4Q9H [61]	3.40 Å	X-ray
	6C0V [62]	3.40 Å	Cryo–EM
	5KPJ [59]	3.50 Å	X-ray
	4XWK [63]	3.50 Å	X-ray
	4Q9L [61]	3.80 Å	X-ray
	4M1M [64]	3.80 Å	X-ray
	5KOI [59]	3.85 Å	X-ray
	6UJN [65]	3.98 Å	X-ray
	5KPI [59]	4.01 Å	X-ray
Sav1866	2HYD [66]	3.00 Å	X-ray
	2ONJ [67]	3.40 Å	X-ray
MsbA	6BPL [68]	2.70 Å	X-ray
	6BPP [68]	2.92 Å	X-ray
	6BL6 [69]	2.80 Å	X-ray
	3B60 [70]	3.70 Å	X-ray
	5TV4 [71]	4.20 Å	Cryo-EM
	6UZL [72]	4.40 Å	Cryo-EM

* Data obtained from the protein data bank (PDB) database (https://www.rcsb.org, accessed on 17 May 2023). Citations to the PDB structures are included. ** Cryo-EM Cryogenic Electron Microscopy; X-ray X-ray Diffraction.

## Data Availability

No new data are reported in this review article.

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
