# Peer review of "The Alternating Access Mechanism in Mammalian Multidrug Resistance Transporters and Their Bacterial Homologs"

_membranes, 2023, doi:10.3390/membranes13060568_

Round 1

Reviewer 1 Report

#135, delete channel because it is a transporter

Author Response

Please see response attached.

Reviewer 2 Report

Shadiand colleagues showed the Alternating Access Mechanism in Mammalian Multidrug Resistance Transporters and Their Bacterial Homologs. The scientific impact of this manuscript is the occurrence of the human community as well in environments, information and awareness to the scientific people. This paper can further help the researcher with as much data related to the Multidrug Resistance (MDR) proteins. I have a few suggestions and queries for the corresponding author regarding the scientific novelty of the manuscript. From my point of view, manuscripts require minor revision or slight modification and allow for a resubmission after that paper is accepted in your journal.

Major:

  1. In this manuscript, authors are not able to pinpoint the definite way by which the MDR protein has some role in resistance or may be role in virulence.
  2. Authors need to add or discuss a particular pathway (machinery) or molecular approach to understand better how the MDR protein has a role.
  3. Also, add or discuss some part of other efflux pump protein.
  4. The abstract needs to be more informative, and as per the journal's high standards, I suggest the author re-frame the whole abstract again. 
  5. The related review section needs to be more robust; no rigorous and extensive related review is found in the article. Authors should include a relevant related review section after the introduction.
  6. The graphical representation of this manuscript is very weak, and the authors need to add some systemic review tables add in this manuscript so that more informative.
  7. Add a few more recent references.
  8. The manuscript should be carefully proof-read regarding the English language.

Author Response

Please see response attached.

Reviewer 3 Report

This manuscript focus in a general overview of ABC transporters, including their classification, structure and mechanism of action, namely regarding their role in the multidrug resistance phenotype. The subject is of great interest and the organization of the paper is good, and in general well written, but some points should be detailed/clarified.

1) When the authors referred the new classification of ABC transporters in seven classes, they should give a brief description why this new classification was implemented and in what is based. Also the english language in the phrase "For instance, group IV (Figure 1) TMD fold with a 6+6 TM helix organization that can be 67 homo- or heterodimer but only in a single chain is where Pgp, MsbA, Sav1866, and MRP1 68 are all categorized [9]." should be revised. In the legend of the figure 1, more details about the structure of this class should be given.

2) The description of the structure, substrate binding site and Alternating Access Mechanism , (AAM) of Pgp, sav1866, MsbA and MRP1 are very well organized and complete. Howeever, why are the bacterial proteins intercalating the human ones? The order should be MRP1 after Pgp and only after the bacteria proteins.

3) The conclusion should be more developed, namely referring the clinical impact that can have the information about these transporters described in this work and highlighting  the new information stated in this work.

Minor aspects in english language can be improved

Author Response

Please see response attached.
